# Electrochemistry of Inorganic OCT-PbS/HDA and OCT-PbS Photosensitizers Thermalized from *Bis*(*N*-diisopropyl-*N*-octyldithiocarbamato) Pb(II) Molecular Precursors

**DOI:** 10.3390/molecules25081919

**Published:** 2020-04-21

**Authors:** Mojeed A. Agoro, Johannes Z. Mbese, Edson L. Meyer

**Affiliations:** 1Department of Chemistry, University of Fort Hare, Private Bag X1314, Alice 5700, South Africa; 2Fort Hare Institute of Technology, University of Fort Hare, Private Bag X1314, Alice 5700, South Africa; emeyer@ufh.ac.za

**Keywords:** semiconductor, molecular precursor, thermal-decomposition, electrochemistry, photovoltaic cells

## Abstract

Inorganic nanocrystal solar cells have been tagged as the next generation of synthesizers that have the potential to break new ground in photovoltaic cells. This synthetic route offers a safe, easy and cost-effective method of achieving the desired material. The present work investigates the synthesis of inorganic PbS sensitizers through a molecular precursor route and their impact on improving the conversion efficiency in photovoltaic cells. PbS photosensitizers were deposited on TiO_2_ by direct deposition, and their structure, morphologies and electrocatalytic properties were examined. The X-ray diffraction (XRD) confirms PbS nanocrystal structure and the atomic force microscopy (AFM) displays the crystalline phase of uniform size and distribution of PbS, indicating compact surface nanoparticles. The electrocatalytic activity by lead sulfide, using *N*-di-isopropyl-*N*-octyldithiocarbamato (OCT) without hexadecylamine (HDA) capping (OCT-PbS) was very low in HI-30 electrolyte, due to its overpotential, while lead sulfide with OCT and HDA-capped (OCT-PbS/HDA) sensitizer exhibited significant electrocatalytic activity with moderate current peaks due to a considerable amount of reversibility. The OCT-PbS sensitizer exhibited a strong resistance interaction with the electrolyte, indicating very poor catalytic activity compared to the OCT-PbS/HDA sensitizer. The values of the open-circuit voltage (*V_OC_*) were ~0.52 V, with a fill factor of 0.33 for OCT-PbS/HDA. The better conversion efficiency displayed by OCT-PbS/HDA is due to its nanoporous nature which improves the device performance and stability.

## 1. Introduction

Research into clean energy has been one of the main concerns of material scientists. The release of environmental pollutants, such as CO_2_, through the use of fossil fuels, has heightened the need for renewable solar energy. The amount of sunlight energy released on a daily basis is more than sufficient if the absorbed photons are completely utilized [1]. Apart from the need for new materials for clean energy, more effort is placed on the synthetic pathways that will deal with the shortfall in the solar cells’ poor conversion efficiency. Limitations, like defects on the nanomaterial surface through impurities emanating from capping agents and lack of control over the particle size, impede composite transport of the charge carriers [1,2]. This has placed a great deal of emphasis on the fabrication process of photovoltaic cells that will result in better conversion efficiency than the ideal dye sensitizer solar cells (DSSC) (see Figure 1), which have been used over the past two decades. The emergence of quantum dots (QDs) sensitizer has found great relevance in the formation of many cells, such as inorganic cells, organic cells, solution-process colloidal solar cells and dye-sensitized solar cells [1,3,4,5,6,7]. Their unique size distribution and spectra properties enable them to absorb the entire visible spectrum, resulting in an improved power conversion efficiency. The use of metals or semiconductors that will produce a smaller bandgap and good optical properties is a major prerequisite.

Inorganic nanocrystal solar cells have been tagged as the next generation of sensitizers that have the potential to break new ground, using semiconductors such as PbS [8]. The adoption of PbS QDs is linked to their near-infrared region properties and excellent photosensitivity, which have made them a better nanomaterial for photovoltaic cells [8,9]. Their direct bandgap of 0.41 eV, high dielectric constant and large exciton Bohr radius (18 nm), have also contributed to their wide application in the production of various items such as photonics, photographic equipment, thermoelectric devices, solar cells and optical fields [10,11,12,13,14]. As diverse applications of PbS QDs in various fields continue to rise, their formation pathways have become a major concern for the material scientist. Various synthetic routes for the fabrication of PbS QDs have been well-documented, such as chemical bath deposition, UV rays, thermal decomposition, microbes and the sonochemical approach [10,15,16,17,18]. These have given rise to various shapes, sizes and morphologies of PbS QDs. Therefore, inorganic nanocrystals are seen as having a major advantage because, with the aid of a few compounds, they offer a safe, easy and cost-effective route to form the desired PbS QD. The interesting chemical properties obtained from the use of dithiocarbamate complexes account for their applications in various fields. In addition to their wide application in the realm of materials science, they offer different arrays of anisotropic and isotropic nanomaterials [19,20,21]. In addition to their wide application, dithiocarbamate complexes offer a clean, nanometric dimension with no impurities when they undergo thermal decomposition [22]. The thermolysis of these metal complexes is one of the easiest and most cost-friendly techniques available for the fabrication of nanoparticles with tunable shapes, minimal defects in structure and narrow size distribution through the aid of capping agents [23,24]. Thermolysis offers various benefits such as controlled conditions, good morphology, purity, environmental friendliness and size distribution [25]. Capping agents, such as hexadecylamine (HDA), trioctylphophine (TOP) and trioctylphophine oxide (TOPO), offer excellent surface passivation, stability and better morphology with organic solvents for nanoparticles [26]. The injection of HDA capping agent provides better particle size and morphology that will enhance the efficiency of the fabricated cells [27]. This work employed molecular precursor techniques to fabricate the photosensitizer with the aid of HDA capping agents, in order to control the structural properties, size distribution and morphologies. Direct deposition was used to coat electrodes with the photosensitizer by immersing them in a solution for a period of time.

## 2. Results and Discussion 

### 2.1. X-Ray Diffraction

The XRD patterns of OCT-PbS/HDA and OCT-PbS photosensitizers are illustrated in Figure 2. Furthermore, the phase structure and purity of OCT-PbS/HDA and OCT-PbS were examined by the XRD analysis. The 2θ values at 19.05, 26.03, 27.03, 34.05, 38.05, 48.07, 52.03, 55.06, 62.04, 66.04, 79.02 and 81.06° are for OCT-PbS. The peak values at 27.03, 31.01, 34.06, 38.05, 52.02, 55.05, 62.05, 66.05, 79.01 and 81.09° are for OCT-PbS/HDA. They both correspond to their crystalline planes of card file No. JCPDC-5-0592 with the diffraction peaks corresponding to (200), (111), (220), (311), (222), (400), (331), (420) and (422) Miller indices. These peaks affirm the PbS QD structure and further confirm the AFM analysis as evidence of successful fabrication of crystalline PbS QDs. The structural disorder emanating from the lattice strain is due to shifts in the diffraction lines. These shifts are typical reflections associated with incorporation into the crystal lattice [28,29,30,31,32]. 

### 2.2. HRTEM

The HRTEM images of OCT-PbS and OCT-PbS/HDA as seen in Figure 3 display crystalline sizes for OCTPbS between 3.16–5.95 nm and OCT-PbS/HDA within 1.82–2.44 nm, with d-spacing of 2.438 and 3.623 nm, respectively. This is consistent with the previous report on d-spacing for PbS [33,34]. The lattice fringe pattern of both materials can be attributed to their polycrystalline nature. The SAED of OCT-PbS and OCT-PbS/HDA indicated a pattern with grain running in various planes, affirming polycrystalline nature typical of the phenomenon of PbS nanoparticles.

### 2.3. Atomic Force Microscopy 

The surface roughness of the OCT-PbS and OCT-PbS/HDA photosensitizers is indicated by the AFM images (see Figure 4). The AFM results for OCT-PbS/HDA indicate that the molecular precursor route with the HDA capping improves the surface roughness: with TiO_2_ photoanode, it is compact and smooth. These results can be linked to the densification and reorganization of the crystallites of the OCT-PbS/HDA film. The photosensitizers exhibited particle size of 0.654 μm for OCT-PbS/HDA and 1.69 μm for OCT-PbS, depicting a regular crystal growth rate. The height clusters of both samples depicted an insignificant change around 0.328 and 1.35 μm, which connotes smaller nanocrystals [35]. The images displayed small spherical nanocrystals with uniform distribution and size in the crystalline phase in both samples, indicating compact nanoparticles [32] and in further agreement with the XRD results. 

### 2.4. Cyclic Voltammetry

In order to evaluate electrocatalytic abilities and the reaction kinetics of PbS/HDA and PbS photosensitizers, the cyclic voltammetry (CV) measurements were adopted with three-electrode systems [36]. The observed CV curves are displayed in Figure 5. In an ideal QDSC, photoexcited electrons from the photosensitizer are injected into the conduction band, and the oxidized photosensitizer is reduced by ions in the electrolyte [37]. The electrocatalytic activity displayed by OCT-PbS/HDA was very low in HI-30 electrolyte due to its overpotential, irreversibility [38] and physisorption. The OCT-PbS sensitizer exhibited significant electrocatalytic activity with stronger current peaks due to a considerable amount of reversibility [39].

### 2.5. Electrochemical Impedance Spectroscopy Results

Figure 6 shows the EIS measurement. The hemisphere of high-frequency at 100 kHz is for the resistance of charge transport at the counter electrode/electrolyte interface (R_1_). At low frequencies, the impedance related to the charge transport at the TiO_2_/PbS/electrolyte interface is R_2_. This study focused only on the R_2_ to compare the effect of the HDA capping agent on the charge transfer and transport at the TiO_2_/PbS/electrolyte interfaces. The impedance at low frequencies can be pinpointed using R_2_. When the R_2_ is lower, the charge transfer is faster at the photosensitzer/electrolyte interface. OCT-PbS/HDA indicated a very poor catalytic activity, which can be linked to the injection of HDA. On the other hand, the OCT-PbS sensitizer displayed lower charge transfer at the photosensitzer/electrolyte interface. This can be linked to the particle size of OCT-PbS sensitizer, which promotes the charge transport speed of the solar cell. This further affirmed the CV and AFM result [40,41]. However, R_2_ is also considered as a resistance of charge recombination at the interfaces of TiO_2_/QDs/electrolytes. Decreases in R_2_ can boost the charge recombination and shorten electron lifetime, which causes the performance of the solar cell to deteriorate.

### 2.6. Bode Plot Results of Metal Sulfides Nanoparticles

According to the EIS diagrams, shown in Figure 7, the electron lifetime before the recombination (*Τ_r_*) can be estimated with Equation (1) using the Bode plot.
[*Τ_r_* = 1/(2*πf*_max_)](1)

The OCT-PbS exhibited a *Τ_r_* value of 57 ms. This indicates an increase in electron lifetime and a backreaction reduction of the electron with the injected HI-30 electrolyte. The suppressing of charge recombination implies the longer lifetime of the charge carrier. This leads to better electron collection at the FTO substrate. Based on this study, we can conclude that OCT-PbS displays superior ability compared to OCT-PbS/HDA. Further, the complete coverage of photosensitizers on a photoanode surface is directly linked to their ability to inhibit the charge recombination back to the electrolyte redox couple [42].

### 2.7. UV−Vis 

Figure 8 reveals the absorption spectra of OCT-PbS and OCT-PbS/HDA recorded in the wavelength region from 315 to 535 nm. This implies a low absorbance in the UV region for OCT-PbS and high absorbance in the visible region for OCT-PbS/HDA. The maximum absorption of OCT-PbS is found at 366 nm, and the absorption is found to be 455 nm for OCT-PbS/HDA. The absorption corresponds to an electron excited by a photon of energy, whereby the electron can jump from a lower energy to a higher energy state. The high absorption displayed by Oct-PbS/HDA in the visible region indicates the blue shift. This may be due to quantum confinement. This enables QDSCs to trap the photon energy in the entire UV-Vis spectral region. Wavelengths similar to those obtained in this study for OCT-PbS/HDA have been reported as produced by the recombination of excitons shallowly trapped in electron–hole pairs by [32,43].

### 2.8. J–V Results

Table 1 and Figure 9 illustrate the current-voltage (*J−V*) characteristics of the QDSSCs of OCT-PbS/HDA and OCT-PbS photosensitizers under illumination. A reasonably high short-circuit current density (*J_SC_*) of 11 mA/cm^2^ was observed in the OCT-PbS/HDA QDSSCs, compared to that of the OCT-PbS cells. The nanoporous nature of OCT-PbS/HDA may be related to the high value of *J_SC_*, due to its fabrication by the single-source precursor technique. The values of the open-circuit voltage (*V_OC_*) (~0.52 V) and efficiency (1.89) for OCT-PbS/HDA proved that the addition of HDA resulted in improved solar cell parameters. The TiO_2_ nanoparticle photoanode, with broad size variation and far-field and near-field effect, led to enhancement of *J_SC_* in OCT-PbS/HDA photosensitizer. This also exhibited high absorption of light in the whole visible spectrum. Secondly, recombination was reduced when PbS nanoparticles functioned as electron scavengers [44,45]. The higher electron density in the TiO_2_ conduction band could be linked as a factor for higher *Voc* value of OCT-PbS/HDA (0.52 V) compared to that of OCT-PbS (0.48 V), which elevated the Fermi level [37]. The FF of QDSSC is usually attributable to the charge transfer resistance at the counter electrode, electron transport resistance through the photoanode, total series resistance of the cell and ion transport resistance [46]. The higher FF of the OCT-PbS (0.74) compared to OCT-PbS/HDA (0.33) was due to the considerable improvement in charge transfer at the counter electrode/electrolyte interface, which reduced the concentration gradients in the electrolyte, internal resistances and the recombination rate, as confirmed by the EIS results [47].

## 3. Materials and Methods 

### 3.1. Material 

All materials were purchased from commercial sources and used without further purification. The complete test kits containing fluorine-doped tin oxide (FTO) as glass substrate of TiO_2_, platinum FTO, HI-30 electrolyte iodide, masks, gaskets, chenodeoxycholic acid (CDC) and a hot seal were purchased from Solaronix Company (Aubonne, Switzerland). Additionally, water, oleic acid (OA), methanol, hexadecylamine (HDA), *bis*(*N*-diisopropyl-*N*-octyldithiocarbamato) (OCT) Pb(II) complexes were obtained from Merck (Johannesburg, South Africa).

### 3.2. Synthesis of OCT-PbS/HDA and OCT-PbS Nanoparticles

Nanoparticles were fabricated according to methods in the literature [48]: 0.20 g of *bis*(*N*-diisopropyl-*N*-octyldithiocarbamato) Pb(II) complex was dissolved in 4 mL oleic acid (OA) and injected into 3 g of hot hexadecylamine (HDA) at 360 °C. An initial temperature of 20–30 °C was attained for the mixture. The reaction was stabilized at 360 °C, and the process lasted for 1 h. The process was allowed to drop to 70 °C, signifying the completion of the process, and about 50 mL of methanol was used to remove excess OA and HDA. Centrifugation was used to separate the flocculent precipitate, which was redispersed with toluene. Low air pressure was used to remove solvent, giving rise to metal sulfides of OCT-PbS/HDA nanoparticles. Synthesis of OCT-PbS nanocrystals was obtained according to the study by a co-worker [49], using (PerkinElmer TGA 4000 thermogravimetric Analyser, San Jose, CA, USA ). A portion of 25 mg of the complex was loaded into an alumina pan and weight changes were recorded as a function of temperature for a 10 °C min^−1^ temperature gradient between 30–900 °C. A purge gas of flowing nitrogen at a rate of 20 mL min^−1^ was used. At temperatures between 360 and 900 °C, the complex end-product was converted into the residue of OCT-PbS nanoparticles from the TGA. 

### 3.3. Fabrication and Assembling of Solar Cells

Quantum dot solar cells (QDSC) were prepared with 2 × 2 cm^2^ FTO-glass plates of platinum and TiO_2_ electrodes, purchased from (Solaronix), with 6 × 6 mm^2^ active areas of TiO_2_ screen-coated. Dye loading for sensitization was done using 10 mL of warm water with OCT-PbS/HDA and OCT-PbS. Co-adsorbents (co-adsorbent/dye), using chenodeoxycholic acid (CDC), were added. The mediating solution was a commercial HI-30 electrolyte solution (Solaronix, Aubonne, Switzerland), with content of iodide species at 0.05 M. The TiO_2_ thin film was soaked in a solution of photosensitizers for 24 h. The two substrates, one coated with TiO_2_ loaded with photosensitizers and the other with platinum, were held together using polyethylene and a soldering iron. A syringe was used to inject the HI-30 electrolyte (iodide).

### 3.4. Physical Measurements

An X-ray diffractometer (Cambridge, United Kingdom) was employed to evaluate the structural pattern of the samples; the diffraction structure results were recorded between 10 to 90° at intervals of 0.05°. A JEOL JEM 2100 high-resolution transmission electron microscope (JEOL Inc., Pleasanton, CA, USA) (HRTEM) operating at 200 KV with selected area electron diffraction (SAED) patterns was used. The surface roughness of the OCT-PbS/HDA and OCT-PbS FTO substrates were identified through the use of atomic force microscopy (JPK NanoWizard II AFM, JPK Instruments, Berlin, Germany) in contact mode and a scan rate of 0.8 Hz. Electrochemical studies were evaluated by Metrohm 85,695 Autolab with Nova 1.10 software (Metrohm Johannesburg, South Africa (Pty) Ltd.). A platinum electrode was adopted as a counter electrode. TiO_2_ was used as the anode, while HI-30 iodide electrode was utilized as a reference electrode. Cyclic voltammetry (CV) was performed at scan rates between 0.05 to 0.35 V s^−1^ with an increment of 0.05 V s^−1^. Electrochemical impedance spectroscopy (EIS) was carried out in the frequency range of 100 kHz to 100 mHz. Current–voltage (*J−V*) parameters were collected through a Keithley 2401 source meter and a Thorax light power meter (RS Components (SA), Johannesburg, South Africa). A Lumixo AM1.5 light simulator was employed, and the lamp was fixed at 50 cm high to avoid illumination outside of the working area. To avoid cell degradation, temperature was kept below 60 °C, and the light power density was kept at 100 mW/cm^−2^ (AM1.5). A PerkinElmer Lambda 25 UV−Vis spectrophotometer was employed to carry out observations of optical absorption properties at room temperature (PerkinElmer, Inc. Waltham, MA, USA). 

## 4. Summary and Conclusions

In summary, the fabrication of inorganic PbS sensitizers through the molecular precursor route and their application in photovoltaic cells were investigated in this study. PbS photosensitizers deposited on TiO_2_ by direct deposition revealed the structure, morphologies and electrocatalytic activity of a typical PbS nanocrystalline structure and displayed uniform size-distribution. The SAED of OCT-PbS and OCT-PbS/HDA confirmed the polycrystalline nature of both materials. The AFM results of OCT-PbS/HDA suggested that the injection of the HDA capping agent improved the surface roughness of the materials with TiO_2_ photoanode. Electrocatalytic activity of OCT-PbS was enhanced in HI-30 electrolyte compared to the OCT-PbS/HDA sensitizer and exhibited significant electrocatalytic activity. The OCT-PbS/HDA sensitizer exhibited a strong resistance interaction with the electrolyte, indicating poor catalytic activity, compared to the OCT-PbS sensitizer. The better conversion efficiency displayed by OCT-PbS/HDA proved its superiority as a good sensitizer, and this is strongly linked to its nanoporous nature and electrocatalytic activity. The addition of an HDA capping agent has made a great improvement to the conversion efficiency of OCT-PbS/HDA. 

## Figures and Tables

**Figure 1 molecules-25-01919-f001:**
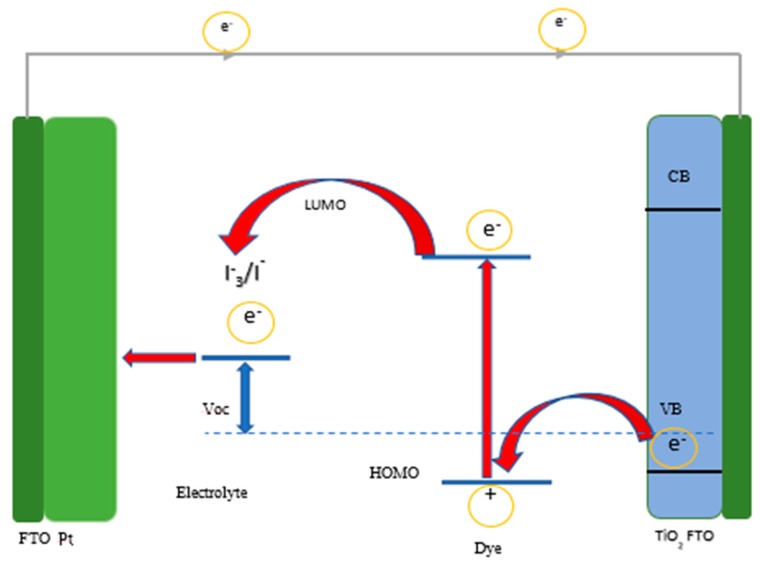
Schematic illustration of ideal dye sensitizer solar cells (DSSC).

**Figure 2 molecules-25-01919-f002:**
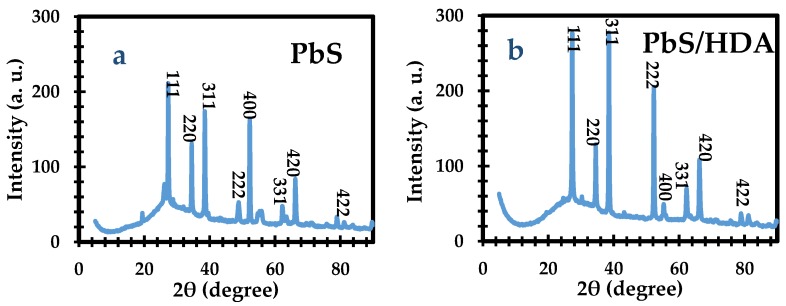
The X-ray diffraction (XRD) spectra of (**a**) OCT-PbS and (**b**) OCT-PbS/HDA nanoparticles.

**Figure 3 molecules-25-01919-f003:**
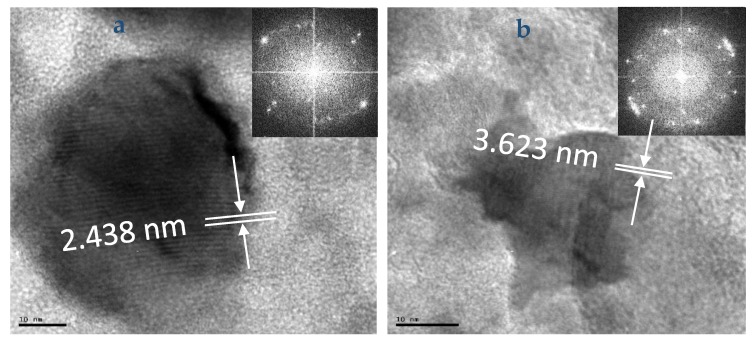
High-resolution transmission electron microscope (HRTEM) images of (**a**) OCT-PBS and (**b**) OCT-PbS/HDA nanoparticles.

**Figure 4 molecules-25-01919-f004:**
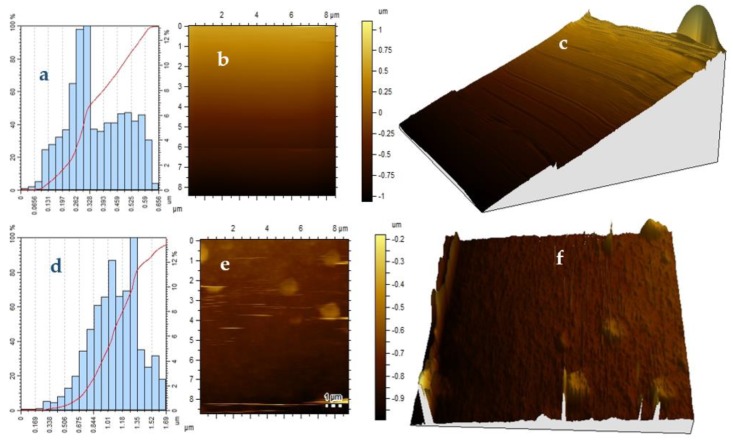
(**a**–**c**) Height profile and two-dimensional (2D) and three-dimensional (3D) atomic force microscopy (AFM) images of OCT-PbS/HAD nanoparticles. (**d**–**f**) Height profile and 2D and 3D AFM images of OCT-PbS nanoparticles.

**Figure 5 molecules-25-01919-f005:**
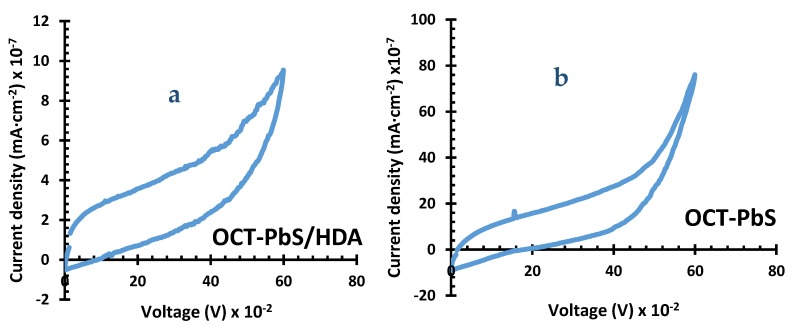
Cyclic voltammetry (CV) spectra of (**a**) OCT-PbS/HDA and (**b**) OCT-PbS nanoparticles.

**Figure 6 molecules-25-01919-f006:**
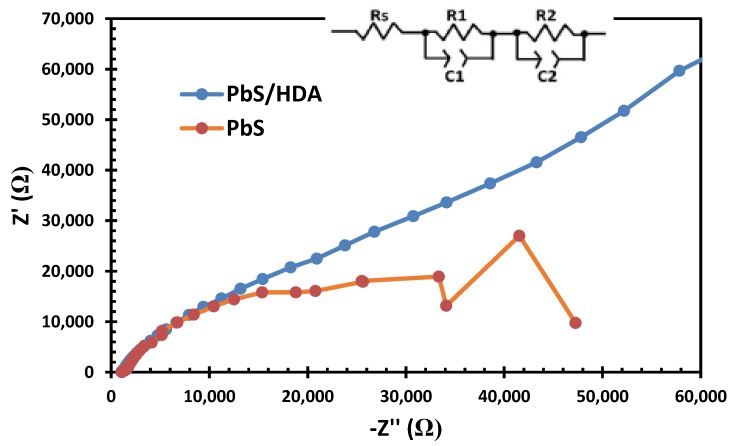
Electrochemical impedance spectroscopy (EIS) spectra of OCT-PbS/HDA and OCT-PbS nanoparticles.

**Figure 7 molecules-25-01919-f007:**
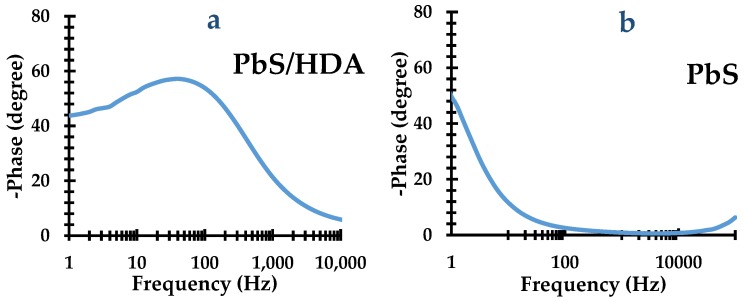
Bode plot spectra of (**a**) OCT-PbS/HDA and (**b**) OCT-PbS nanoparticles.

**Figure 8 molecules-25-01919-f008:**
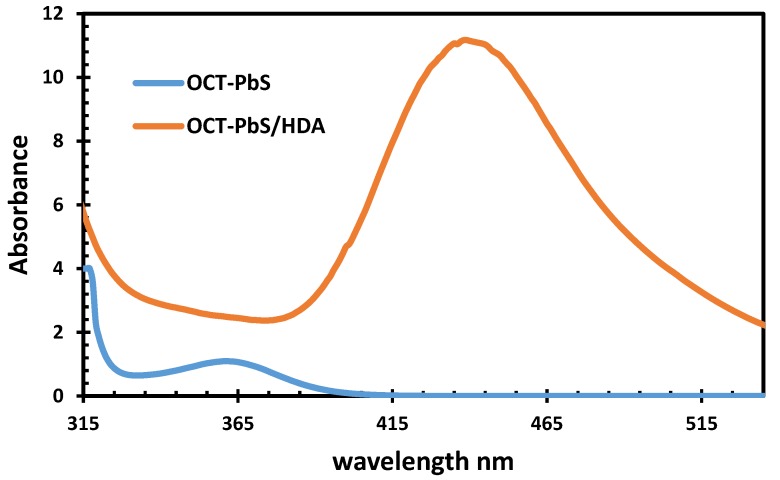
UV−Vis image of OCT-PBS and OCT-PbS/HDA nanoparticles.

**Figure 9 molecules-25-01919-f009:**
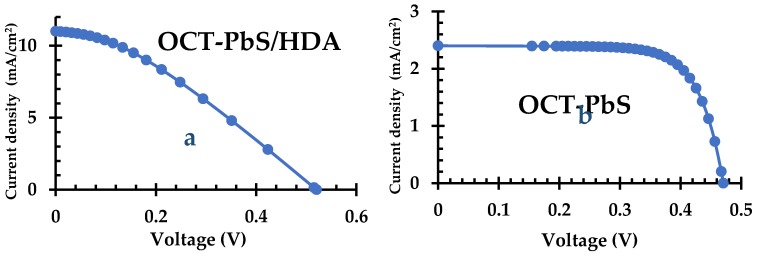
*J−V* curve characteristics of (**a**) OCT-PbS/HDA and (**b**) OCT-PbS nanoparticles.

**Table 1 molecules-25-01919-t001:** Current-voltage (*J−V*) curve characteristics of OCT-PbS/HDA and OCT-PbS nanoparticles.

Dye	Photoanode	Electrolyte	CEs	*J_SC_* (mA/cm^2^)	*V_OC_* (V)	*FF*	*η* (%)
**OCT-PbS/HDA**	TiO_2_	HI-30	Pt	11	0.52	0.33	**1.89**
**OCT-PbS**	TiO_2_	HI-30	Pt	2.4	0.48	0.74	**0.85**

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
