# Peer review of "Electrochemistry of Inorganic OCT-PbS/HDA and OCT-PbS Photosensitizers Thermalized from Bis(N-diisopropyl-N-octyldithiocarbamato) Pb(II) Molecular Precursors"

_molecules, 2020, doi:10.3390/molecules25081919_

Round 1

Reviewer 1 Report

This manuscript showed a little improvement after resubmission. But there are still many problems in the manuscript. It is suggested to be reconsidered after major revision.

  1. The authors should check the unit through the whole manuscript. There are many unit errors. For example, the unit of Voc should be V but the authors used mV for many times, even in the abstract. And in many units, the numbers should be superscript but not.
  2. In Table 1, the Voc and FF of OCT-PbS based solar cell is with too many scientific digits.
  3. The authors did not define OCT in the manuscript.
  4. In Figure 3, the d-spacing is not very clear labeled, especially for the OCT-PbS/HAD.
  5. The discussion of the EIS measurement (above Figure 6) is not clear. The authors did not define the “Rct” before they use it. And the language of the paragraph of the discussion above Figure 6 is not very understandable, as it contains many grammar errors and typos. The authors should polish this part.
  6. The discussion of the performance of solar cell is not enough. The authors basically explain the Jsc difference between OCT-PbS and OCT-PbS/HAD based devices. But the difference of Voc and FF are not much touched. Especially, the difference of FF between OCT-PbS and OCT-PbS/HAD based devices is very large. The authors should give an explanation.

Author Response

Please find the attached response to the reviewer 1 comments for manuscript ID number 761103

Reviewer 2 Report

The author has revised the manuscript according to the referee's suggestion.  

Author Response

The reviewer 2 comments for manuscript ID number 761103 indicated that the authors has revised the manuscript according to the referees suggestion.  

Round 2

Reviewer 1 Report

This manuscript has been revised several times, but it still contains some problems and errors. It is suggested to be reconsidered after major revision.

  1. The authors attribute the low FF in OCT-PbS based solar cell to “the considerable improvement in charge transfer at the counter electrode/electrolyte interface” compared with OCT-PbS/HDA based solar cell. The counter electrode is platinum electrode. And they used EIS measurement as evidence. But in the discussion of EIS measurement, the authors claimed charge transfer impedance is for OCT-PbS/electrolyte interface transport. The authors used EIS measurement for two different interfaces. The author should give a good explain.
  2. In the explanation of improved Jsc in OCT-PbS/HDA based solar cell, the authors claimed, “This also exhibited high absorption of light in the whole visible spectrum.” But there is no evidence to support the high absorption. The authors should give the absorption spectrum.
  3. In Figure 5, the description of the CV looks opposite to the curve shown. The authors claimed “The electrocatalytic activity displaced by OCT-PbS were ver low in HI-30 electrolyte” and “OCT-PbS/HDA sensitizer exhibited significant electrocatalytic activity with moderate current peaks” but in the figure OCT-PbS showed stronger current peak.
  4. Unit errors still exist in the manuscript. For example, in Figure 8, the unit of current density is not right and in Figure 5b the unit of voltage is not right as well.
  5. In the XRD, the 2theta of OCT-PbS only goes to 80 degrees in Figure 2a. But in the main text, there is a peak at 81.06 degree from OCT-PbS, which is out of the range of 2theta in the figure.
  6. The language requires polish. The authors used incomplete sentences for many times. For example, “Thereby, promoting the charge collection of high current density and increasing the charge transport speed of solar cell” and “The values of the open-circuit voltage (VOC) (~0.52 V) and fill factor (0.33) for OCT-PbS/HDA.”

Author Response

Please find the attached document for molecules ID number 761103 response to reviewer 1 round 2 comments

This manuscript is a resubmission of an earlier submission. The following is a list of the peer review reports and author responses from that submission.

Round 1

Reviewer 1 Report

The present work describe the fabrication of inorganic PbS sensitizers through molecular precursor route and there application in photovoltaic cells. The manuscript presents an elements of novalty, however some points should be better presented. A minor revision is recommended based on the following considerations: 1) Ther authors should evaluate the electrocatalytic activity of the materials using the I/I3 system; 2) The authors should perform a new EIS measurement for the Figure 4B since the spectrum presented in this figure shows a high dispersion of data. Please check the reference electrode and the electric contacts when you perform the new mesurement; 4) A schematic diagram of the work principle of DSSC should be presented.

Reviewer 2 Report

In this work, the authors present a synthetic method of inorganic PbS sensitizers through molecular precursor route and there impart in improving the conversion efficiency in photovoltaic cells. Compared to OCT-PbS sensitizer, OCT-PbS/HDA has a good conversion efficiency in HI-30 electrolyte and is a sensitizer with strong electrocatalytic activity. It can be accepted after some supplementary experiments and modifications.

  1. It would be better if the author can provide the SEM or TEM image of OCT-PbS/HDA and OCT-PbS photosensitizers, in order to show the morphology and uniformity.
  2. For the Figure 2, author may have to increase font sizes for legibility.
  3. For the Figure 3 in the paper, it will be more concise and consistent if the numerical value in y axis could be converted to exponent form.
  4. The author should pay attention on the mistake in the paper, for example, in page 2, line 70 ‘Water, Oleic acid (OA), methanol, hexadecylamine (HDA), OCT-PbS/HDA and OCT-PbS nanoparticles from bis(N-di-isopropyl-N-octyldithiocarbamato) Pb(II) complexes.’

Reviewer 3 Report

This manuscript did the study of synthesis route of PbS QDs in Inorganic nanocrystals solar cells. The authors used HDA to modify the OCT-PbS. The modified OCT-PbS/HDA showed better performance than the OCT-PbS based solar cell.  

  • The manuscript is not well organized. The authors do not show the purpose and novelty of the manuscript in introduction. The background of similar synthesis modification work is not introduced as well. 
  • The figures contain many problems, such as the unit mistake in y axis in Figure 3, the number in x axis in Figure 1a not properly labeled, the y axis in Figure 6 mislabeled from current density to current and the low quality of AFM image in Figure 2e and 3D image in Figure 2f.
  • The important data and statement are not consistent in the manuscript. The authors used a AM 1.5 solar simulator to measure under AM 1.0 condition. In Figure 3 the CV curves are not start from zero. In Figure 2, the AFM images of OCT-PbS and OCT-PbS/HAD are not in the same size or area. Then the comparison is not fair. Most importantly, the J-V curve of OCT-PbS/HDA in Figure 6a is completely different from the data listed in Table 1. Both the Voc and FF is not right. Then the conclusion based on the comparison of those data is not acceptable. It is suggested to be rejected.

Reviewer 4 Report

The submitted manuscript presents Electrochemistry of inorganic PbS photosensitizer by using Pb(II) molecular precursors. The morphology and physical properties have been investigated by various method; The I-V curve characteristics of OCT-PbS/HDA and PbS nanoparticles are compared. However, some questions should be addressed for the clarify of the work.

Main questions:

  1. Line 13~line 27, the full name of the abbreviations of words such as OCT, HAD, AFM should be given as you use them first time. Please check all these problems in the whole manuscript.
  2. Quantum dots have been well developed for photovoltaic application. Reference 3-7 is not adequate. The following references should be cited.

 https://doi.org/10.3390/app9091885, https://doi.org/10.1016/j.nanoen.2018.01.048, https://doi.org/10.1002/adfm.201904018, https://www.mdpi.com/2079-4991/9/4/626

  1. Figure 1 (a), OCT-PBS should be written as OCT-PbS.
  2. Figure 2, Please write out the preparation process of the OCT-PbS and OCT-PbS/HDA film in detail.
  3. Note that all physical units must be shown in italics. For example, the Jsc should be Jsc, Voc should be Voc

Round 2

Reviewer 3 Report

I did not feel much improvement for this manuscript. It is suggested to be rejected. 

1. The authors still did not give good introduction. They did not explain why they used HDA to modify the OCT-PbS. What is the advantage of HDA compared with other chemicals, which can modify the OCT-PbS? Those should be clearly mentioned in the introduction to show the novelty of this manuscript.

2. The size of two AFM images is still different (one is 8um by 8um, the other is 3.5um by 3.5um). The authors compare two AFM image with different size. I do not think they are comparable. The authors should scan in the same size to compare. Otherwise, it is not a fair comparison. And the AFM image of OCT-PbS/HDA is still in low quality, the image looks like the AFM tip had too strong force and damaged the sample surface as lines showed in the AFM image. The authors should redo the AFM. 

3. The efficiency parameters listed in Table 1 are still not consistent with the J-V curve. The authors changed the FF value but they did not realize the final efficiency is wrong now, it should be only about 3.2%. And even the new FF value is still suspicious. The J-V curve in Figure 8a shows a bad FF, but the authors give a FF value of 0.56. I really doubt the accuracy of the data. The author should supply the raw J-V curve data to verify their efficiency. 

BTW, the unit of x axis and y axis in Figure 8 is wrong. And the caption of Figure 8 should be "J-V curve" not "I-V curve". And the unit of Voc is mV in Table 1 but should be V. 

4. The authors did not change the wrong values of FF and Voc in abstract.

5. The authors mentioned they used an "AM 1.5" light simulator on Page 3, line 112, but they said the measurement was taken in "AM 1.0" condition in line 114. Is it a typo?